# The International Competition on Knowledge Engineering for Planning and Scheduling: Food for Thoughts (and Call to Action)

**Mauro Vallati**
School of Computing and Engineering
University of Huddersfield

**Lukáš Chrpa**
Faculty of Electrical Engineering
Czech Technical University in Prague &
Faculty of Mathematics and Physics
Charles University in Prague

## Abstract

The International Competition on Knowledge Engineering for Planning and Scheduling (ICKEPS) plays a pivotal role in fostering the development of new Knowledge Engineering (KE) tools, and in emphasising the importance of principled approaches for all the different KE aspects that are needed for the successful long-term use of planning in real-world applications. In this paper, as an exercise in synthesis and for the sake of stimulating thoughts and discussion, we review the format of previous ICKEPS, to suggest alternative formats for future competitions, ideally to motivate someone to step up and organise the next ones.

## Introduction

The International Competition on Knowledge Engineering for Planning and Scheduling (ICKEPS) has been running since 2005 as an almost biennial event promoting the development and importance of the use of knowledge engineering (KE) methods and techniques within this area. The aim of the competition series is to foster developments in the knowledge-based and domain modelling aspects of Automated Planning, to accelerate knowledge engineering research, to encourage the creation and sharing of prototype tools and software platforms that promise more rapid, accessible, and effective ways to construct reliable and efficient Automated Planning systems.

The latest competition took place in 2016[1] (Chrpa et al. 2017), which aimed at on-site domain modelling, and highlighted a number of major issues. Most teams did not use any of the existing KE tools, and thus relied only on their expertise. Second, existing tools do not effectively support cooperation, which is needed to cope with the growing complexity of planning applications. Finally, and more worryingly, the number of participants of ICKEPS is still not very large, especially when compared with the latest edition of the International Planning Competition: this suggests that the planning community underestimates the importance of knowledge engineering, despite of its enormous impact on applicability of domain-independent planning in real-world scenarios. Accidental complexity issues (Brooks 1987), for instance, can prevent the exploitation of automated planning

---

[1]Detailed information can be found at http://ickeps2016.wordpress.com/

approaches in complex scenarios, and even an unfortunate ordering of elements in the domain model can adversely affect the performance of planning engines (Vallati et al. 2015).

Given the pivotal role played by ICKEPS in promoting the importance of principled KE approaches and tools, we believe it is important to evolve and adapt its format in order to attract and engage a larger number of participants. In this paper, we review the format of past competitions, in order to highlight weaknesses and strengths both from organisers' and participants' perspective. Building on top of this analysis, we suggest some alternative formats that may help future ICKEPS organisers in performing their tasks.

It should be noted, though, that the aim of this paper is twofold: to review formats and suggest improvements to ICKEPS, and –more importantly– to make a call for action for organising future competitions focused on KE aspects of planning and scheduling.

## Formats of ICKEPS

This section is devoted to describe the formats of past ICKEPS.

### General Tools Design

The first edition of ICKEPS, held in 2005 (Barták and McCluskey 2006), focused on tools for KE. Any tool that helped in knowledge formulation (the acquisition and encoding of domain structure or control heuristics), planner configuration (fusing application knowledge with a Planning or Scheduling engine), validation of the domain model (for example, using visualisation, analysis, reformulation) or validation and maintenance of the Planning and Scheduling system as a whole (for example, using plan/schedule visualisation, or automated knowledge refinement) was allowed to take part.

The competition included two stages. In the pre-competition stage, the competitors submitted short papers describing the tools. The program committee did light reviewing of the papers with the goal to evaluate relevance of the tools, to send feedback to the competitors, and to contribute to the overall evaluation. During the on-site competition, the participants gave talks about their systems in a workshop-like arrangement, and then they presented the systems during an open demonstration session.

**Evaluation**  The tools were evaluated by a jury of experts against the following criteria[2]:

- support potential: what potential has the tool in helping the processes within the scope of the competition? Will the tool save time and resources?

- scope: how broad is the scope of the tool within the defined scope of the competition?

- usability: can the tool be easily used, accessed and/or configured? Could non planning-experts use it?

- interoperability: can the tool be integrated with other Planning and Scheduling technology? Are its interfaces well defined - can the software be easily used with other Planning and Scheduling software, or easily combined with third party planners?

- innovation: what is the quality of the scientific and technical innovations that underlie the software?

- wider comparison: How does the tool compare with KE software in other areas of AI? For example, could the software be subsumed by some other existing Knowledge-Based system KE tool?

- build quality: does the software appear robust? Has the software been well tested?

- relevance: to what degree does the tool address problems typical to KE for Planning and Scheduling? Is the software relevant or applicable to real-world applications?

## General Tools Design and Simulation

The 2007 edition of ICKEPS[3] extended the above format by including an additional simulation stage. A web service including a number of planning and scheduling simulations was made available to participants, in the pre-competition stage, to evaluate their tools. Competitors were made available a short text description of the competition domain, including a description of the simulation API. They used their tools to encode models and submit generated plans for each instance, and received feedback describing the quality of the plan.

**Evaluation**  As in ICKEPS 2005, tools were evaluated by judges by taking into account a number of criteria. In 2007, above mentioned criteria were extended by considering also aspects related to the simulation:

- domain simulation applicability: how well did the competitors address the simulation domains using their tools? How many domains were the simulators tried on? How long did it take competitors to generate valid plans for the domains? How many problem instances were solved? What was the quality of the plans generated?

## Specific Tools Design

ICKEPS 2009 (Barták, Fratini, and McCluskey 2010) exploited the same format of ICKEPS 2005, based on papers' submissions and workshop-like demonstrations, but narrowed the scope to tools that support a specific aspect of knowledge engineering technology: those that when input with a model described in an application-area-specific language, output solver-ready domain models. The rationale was to foster the development of tools that can support a rapid exploitation of automated planning in real-world applications, by leveraging on existing planning engines.

**Evaluation**  Evaluation was performed by a board of judges that considered a wide range of criteria, divided into two main classes: criteria focusing on the *software engineering aspects* of the tools (e.g., robustness, usability, etc.), and criteria focusing on the more traditional Planning and Scheduling elements, such as originality, comprehensiveness, etc.

## Off-site Modelling and Demonstration

The 2012 edition of ICKEPS[4] included two different tracks: The Design Process Track, and the Challenge Track. The former followed the structure of previous ICKEPS, and was focused on the design of both general and specific tools for KE.

The newly-introduced challenge track aimed at evaluating the actual usefulness of tools and approaches in tackling complex application domains of Planning and Scheduling. Participants were provided a few months before the actual competition with the natural-language specifications of 3 challenging scenarios for planning and scheduling, and had to tackle one off-site. During the workshop-like demonstration, the participants had to demonstrate the advantage of using their tools/method to produce a model as a solution to the requirements (or a sub-set of) in the specification and the plans for the specified scenarios.

The evaluation criteria were the same used in ICKEPS 2009.

## On-site Modelling and Demonstration

This format was introduced in ICKEPS 2016, and included two main stages: on-site modelling and a subsequent demonstration.

During the first stage, each team received descriptions of 4 scenarios and had to exploit the available time for generating the corresponding models. Scenarios were not taken from real-world applications of planning, but were designed by the organisers taking inspiration from games or from potential application domains. Participants were free to select the scenarios to tackle, and had no restrictions on the number and type of tools that can be used. The only constraints were on the available time –six hours were given– and on the maximum size of teams: at most four members. The day after, each team had to present, in a 10-minute demonstration, the aspects of the knowledge engineering process they exploited for encoding the scenarios. Specifically, teams were expected to discuss: the division of work among team members, the tools used, key decisions taken during the encoding, and the issues they faced.

---

[2]http://idm-lab.org/wiki/icaps/ickeps2005/rules.html

[3]http://idm-lab.org/wiki/icaps/ickeps2007/

[4]http://icaps12.icaps-conference.org/ickeps.html

| Format | Pros | Cons |
|---|---|---|
| Tool design | Provide the community with tools | Tools can be hard to compare. Design and development are very hard and can discourage participants. May be hard to develop something innovative. |
| Off-site modelling | Possible to consider challenging cases. Time to exploit principled KE approaches. | Hard to identify suitable domains and models. |
| On-site modelling | More attractive. Can allow to distil good practice in KE. | No new tools for the community. Only toy domains can be considered. |

Table 1: Overview of strengths and weaknesses of considered ICKEPS formats.

**Evaluation** Evaluation included both qualitative and quantitative aspects, and focused on three aspects:

- KE tools exploited. This included the list of tools, the KE steps covered by the tools, etc.

- Model characteristics. Models were checked in terms of presence bugs, number of operators, readability, etc.

- Observed planners' performance. Encoded models were tested using a set of planners, in order to extract useful statistics to be used to empirically compare models.

The jury of experts was present at the demonstration, and took into account the above mentioned aspects to award the teams that excelled in all (or in some) of the aspects.

## Strengths and Weaknesses of ICKEPS Formats

In this section we highlight strengths and weaknesses of the format of past ICKEPS, with the aim of synthesising some suggestions for future competitions. An overview of this analysis is provided in Table 1.

ICKEPS based on tool design (either specific or general) have fostered the development of a decent number of KE tools, that are exploited by the wider planning community and are extremely helpful for testing planning in real-world applications. The main issue of this format of competition is that, nowadays, quite a significant number of tools is available, and it is hard to provide innovative general tools. Furthermore, the design and development of such tools and techniques in exceedingly demanding, and this has a strong impact on the number of competitors. In the case of general tools, the comparison can also be cumbersome: tools that can support the formulation of models in different languages, or that are aimed at supporting different aspects of KE for planning can be extremely hard to compare. On the other hand, this sort of competitions can be very useful when the focus is on specific aspects of KE / specific languages. This focus can help in the comparison, and can foster the work on some overlooked areas of KE, but may also significantly limit the interest of the community and the number of participants.

Off-site modelling and demonstration poses significant burden to the organisers, because they have to identify a set of application domains, or a specific angle within some previously explored domain, where automated planning has not been applied before, and where it is possible to create models that are challenging from a formulation point of view, and at the same time can allow domain-independent planning engines to generate solutions in a reasonable amount of CPU-time. On the other hand, the large amount of time made available to competitors allows to consider more complex cases than those that can be handled in an on-site competition. Furthermore, the more relaxed settings also fosters the use of principled knowledge engineering approaches and techniques.

Intuitively, the on-site modelling format has pros and cons which are quite the opposite of what has been discussed for off-site modelling competitions. The limited amount of time available to participants for formulating domain models forces the organisers to consider only "toy" examples, and does not allow to push the boundaries of planning in real-world applications. As observed in ICKEPS 2016, it is usually the case that models are so easy that no KE tool is needed, but a text editor is enough to encode reasonably good models. On the plus side, this style of competition, that is inspired by Hackatons and similar events, lead to a more "funny" sort of competition, and can attract also people that are not usually interested in KE aspects of planning. This is particularly true for students and young researchers. Furthermore, an analysis of techniques exploited by participants can also lead to identify good practice in KE, that can be emphasised and discussed during the demo session or after the competition.

## Ideas for Future ICKEPS Format

Distilling the knowledge obtained by reviewing the formats of past ICKEPS, we found ourselves in the position of suggesting two possible structures for future competitions, that may help to keep alive the interest of the wider ICAPS community in KE aspects, and provide useful data or tools as a tangible heritage.

It may be worth to revive formats focusing on what we previously defined as Specific Tool Design. Past ICKEPS exploiting this format considered tools able to translate models between different languages. Taking into account different areas may help to foster the development of tools, and also can help providing some sort of standards that can be used for future work in the area. An example area could be, given also the special attention given to the topic by the 2019 workshop on Knowledge Engineering for Planning and

Scheduling (KEPS)[5] (McCluskey et al. 2003), on automated domain model acquisition. With Specific Tool Design focus, it might be possible to specify expected type and amount of input information as well as expected output. Consequently, it might be possible to specify metrics that can be used to quantitatively evaluate a particular tool. Notably, the metrics might follow (soft) constraints that can be set according to a specific application domain, where only some type of data can be expected, or according to some more general "usual" conditions. Such a format can motivate development of tools that might be of a critical importance for advancing the state-of-the-art in the area of KE for Planning and Scheduling. Moreover, with the quantifying metrics the tools can be evaluated more objectively and thus mitigating subjective assessment of judges. Also, to some extent such a format should reduce burden of the organisers as they might focus on narrower scope of the competition.

Another suitable format for future ICKEPS can be obtained by mixing off-site and on-site modelling, aiming at exploiting the strengths of both. A suitable combination may be the following: participants are provided with the specifications of "not so easy" application domains to model in a planning language (e.g. PDDL), and demonstrate that existing domain-independent planning engines can handle given sample planning instances from the domains and provide good quality solutions in a reasonable amount of CPU-time. In the on-line stage of the competition, participants will be provided with a modified version of the specifications, and will be required to modify the models accordingly. Again, the domain models will be evaluated on given samples of planning instances. The intuition behind the idea is to foster the exploitation of principled KE approaches for formulating domain models, especially in the off-site stage, with a focus on robustness and maintenance of the models, and possibly help shaping a notion of quality of domain models (McCluskey, Vaquero, and Vallati 2017), such that the model amendment in the on-site stage would be manageable. The potential issue for the organising team is find appropriate and interesting application domains that are both reasonable challenging to formulate, and suitable to "not so easy or hard" modifications.

## Conclusion

Concluding this paper, we believe that there is a strong need to organise the ICKEPS competitions in order to increase awareness of KE techniques, tool and issues in the ICAPS and general AI communities. The success of future ICKEPS competitions (e.g. considerable increase of the number of participants) can, in consequence, influence the domain-independent AI planning field by making it accessible for use (by planning non-experts) in various application domains. To give some motivation and inspiration for the future ICKEPS competitions, we, in this paper, provided a review of the format of the past ICKEPS competitions, and suggested two possibly new formats that, we believe, can at-

tract more participants and possibly avoid an excessive burden of organisers.

We believe that the paper initiates a fruitful discussion about the format of future ICKEPS competitions as well as motivate potential organisers to step up and organise the next competition(s).

## Acknowledgements

This research was funded by the Czech Science Foundation (project no. 18-07252S).

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
