# OpenReview forum: "The International Competition on Knowledge Engineering for Planning and Scheduling: Food for Thoughts (and Call to Action)"
_icaps-conference.org/ICAPS/2019/Workshop/KEPS — KEPS 2019_

### Official Review · AnonReviewer1 · 2019-05-13
**Short paper with significant insights and discussion**

**Rating:** 5
**Confidence:** 3

**Review:**

This short paper provides a rather interesting analysis and discussion over insights after the organization of several ICKEPS editions.

The paper identify pros and cons of past editions and propose some possible options for the future with the aim of increasing the number of participants in the competition and, hopefully, leverage its results for advancing the state of the art.

The paper seems to me very well suited for KEPS as it can really  foster an highly interesting discussion among its potential attendees.


KE (applied to P&S) is a rather wide field and it seems to me hard to define "bounds" within which a tool/solution/methodology can be evaluated in a quantitative way. As discussed by authors, going in that direction entails several issues that may lead to not significant challenges (e.g., using toy problems) or very difficult assessments.
Also restricting the competition to PDDL does not seem an exciting perspective to me.

My personal opinion is that the organizers of future ICKEPS editions should look back at the first editions considering sort of open challenges tracks in which researchers can provide wider kind of solutions and be more free to investigate actual challenges. I would then encourage to look for something really interesting (and coming from real applications) rather than building artificial problems. Indeed, the risk is to build wonderful KE tools capable of dealing artificial problems/situations but that nobody cares about...or having something similar to IPC, i.e., planners very good/effective in solving (toy) planning problems but with (many) difficulties in addressing real world scenarios.

---

### Official Review · AnonReviewer2 · 2019-05-16
**This is a triggering paper to request idea and (also) volunteering for help organizing the next ICKEPS.**

**Rating:** 4
**Confidence:** 2

**Review:**

the paper does the job: fostering interest around a new run of the KE competition.
being myself one of the supporters of the KE task I do really support the talk behind this paper and its effort to revive elaboration around the topic.
One additional comment: I would vote for not abandoning the direction of promoting the use of tools that facilitate KE.